# HIGH-AVATAR: HIERARCHICAL REPRESENTATION FOR ONE-SHOT GAUSSIAN HEAD AVATAR

## ABSTRACT

We propose HIGH-Avatar, a novel one-shot method that leverages a **HI**erarchical representation for animatable 3D **G**aussian **H**ead reconstruction from a single image. In contrast to existing approaches with a fixed number of Gaussians, our method enables multi-LOD (Level-of-Detail) head avatar modeling using a unified model. To capture both global and local facial characteristics, we employ a transformer-based architecture for global feature extraction and projection-based sampling for local feature acquisition. These features are effectively fused under the guidance of a depth buffer, ensuring occlusion plausibility. A coarse-to-fine learning strategy is introduced to enhance training stability and improve the perception of hierarchical details. To address the limitations of 3DMMs in modeling non-head regions such as the shoulders, we introduce a multi-region decomposition scheme, where the head and shoulders are predicted separately and then integrated through cross-region combination. Extensive experiments demonstrate that HIGH-Avatar outperforms state-of-the-art methods in terms of reconstruction quality, reenactment performance, and computational efficiency.

## 1 INTRODUCTION

Reconstructing an animatable 3D head avatar from a single image is a crucial and rapidly evolving research area in computer vision and graphics. This technology has great potential for applications across various domains, including the game and video production industries, virtual meetings, and the emerging Metaverse. To facilitate the widespread adoption of this technology, several key features are essential: high-efficiency reconstruction and inference, rich facial details, and precise controllability over expressions and head poses. In recent years, numerous methods have been developed to tackle this task, which can be divided into 2D-based and 3D-based approaches.

Early 2D-based methods (Siarohin et al., 2019; Wang et al., 2021; Guo et al., 2024) predict deformation flows to warp the latent features of a source portrait and employ GANs (Generative Adversarial Networks) (Goodfellow et al., 2014) to synthesize the reenacted output. With the rise of latent diffusion models (Rombach et al., 2022), recent approaches (Tian et al., 2024; Jiang et al., 2024) have adopted cross-attention mechanisms conditioned on driving signals, achieving superior image quality and better appearance preservation. However, both GAN-based and diffusion-based methods require substantial computational resources, limiting their applicability in real-time scenarios. Furthermore, due to the lack of 3D constraints, these approaches often struggle to maintain multi-view consistency under large pose or viewpoint variations.

In the 3D avatar domain, NeRF and Gaussian Splatting have emerged as prominent approaches due to their high-quality representation and rendering capabilities. Compared to NeRF-based methods (Bai et al., 2023a; Gafni et al., 2021; Ki et al., 2024; Li et al., 2023a; Ma et al., 2023; Park et al., 2021a; Yu et al., 2023; Zheng et al., 2023), 3D Gaussian Splatting has become the prevailing choice owing to its significantly fast rendering speed. Unlike early Gaussian-based approaches (Xu et al., 2024b; Wu et al., 2024; Tang et al., 2025) that require extensive per-individual optimization, recent methods like GAGAvatar (Chu & Harada, 2024) and LAM (He et al., 2025) propose a one-shot head avatar reconstruction framework, improving generalization capabilities. Despite these advancements, key challenges persist in scalability, efficiency, and modeling completeness. For instance, GAGAvatar primarily utilizes Gaussian points sampled from the image plane, which makes it less of a true 3D head model. Additionally, many of these Gaussian points correspond

to background regions, leading to redundancy and inefficiency. Conversely, LAM adopts subdivided FLAME vertices as queries for image features extraction via a cross-attention mechanism, but its computational complexity grows exponentially with subdivision times, limiting its scalability. Moreover, its performance degrades sharply as the number of points decreases, making it unsuitable for level-of-detail (LOD) rendering. Lastly, the reliance on the FLAME head model constrains its representation ability for non-head areas, such as shoulders.

To address these challenges, we propose a novel one-shot method for generating 3D Gaussian head avatars with hierarchical representation. Rather than performing 2D-to-3D feature mapping on high-resolution meshes, which is computationally expensive, our approach extracts both global and local features from low-resolution meshes and progressively refines them to high-resolution representations through subdivision operations during the training phase, this also enables dynamic multi-LOD rendering at runtime. By leveraging an effective occlusion-aware feature fusion mechanism, our model delivers superior reconstruction quality while significantly reducing computational cost and the number of Gaussians compared to existing methods. Additionally, to improve representation in non-head regions, we independently model the head and shoulders based on shared features, greatly improving the completeness of the generated avatars.

The main contributions of our work are summarized as follows:

- We present a novel hierarchical framework for one-shot Gaussian head modeling, enabling dynamic multi-LOD rendering while achieving superior reconstruction quality and inference speed with significantly reduced computational cost and less number of Gaussians.

- To better capture facial details and improve training efficiency, we propose a coarse-to-fine learning strategy that progressively refines both transformer-based global features and projection-sampled local features through multi-level subdivisions, followed by an occlusion-aware feature fusion mechanism guided by the depth buffer.

- We propose a multi-region decomposition scheme to model heads and shoulders separately, significantly enhancing the fidelity and completeness of generated avatars.

## 2 RELATED WORK

**2D Talking Head Generation** Early 2D approaches (Zakharov et al., 2019; Burkov et al., 2020; Zhou et al., 2021; Wang et al., 2023) employ generative adversarial networks (GANs) (Goodfellow et al., 2014; Isola et al., 2017; Karras et al., 2020) and incorporate driving expression features for controllable portrait synthesis. Subsequent methods (Siarohin et al., 2019; Ren et al., 2021; Drobyshev et al., 2022; Hong et al., 2022; Zhang et al., 2023; Guo et al., 2024) adopt deformation-based frameworks, representing expressions and poses as warping fields to deform the source image. Recent diffusion-based approaches (Cui et al., 2024; Tian et al., 2024; Xu et al., 2024a) further improve visual quality and temporal coherence, but their high computational cost limits real-time performance. Despite these advances, 2D methods struggle with large pose and expression variations due to a lack of 3D awareness. To address this, some (Nef, 1999; Paysan et al., 2009; Li et al., 2017; Gerig et al., 2018) integrate 3D Morphable Models (3DMMs) into 2D pipelines, but they still lack support for viewpoint control and free-viewpoint rendering.

**3D Head Avatar Generation** Traditional 3D head avatars typically rely on 3D Morphable Models (3DMMs) for mesh reconstruction (Xu et al., 2020; Khakhulin et al., 2022), which often fail to capture fine geometric details. In contrast, NeRF-based approaches (Kirschstein et al., 2023; Athar et al., 2022; Bai et al., 2023b; Gafni et al., 2021; Gao et al., 2022; Guo et al., 2021; Ki et al., 2024; Park et al., 2021a;b; Tretschk et al., 2021; Zhang et al., 2024; Zhao et al., 2023; Zheng et al., 2023; Zielonka et al., 2023) have significantly improved reconstruction accuracy and detail representation, and several efficient one-shot NeRF methods have also been proposed (Yu et al., 2023; Li et al., 2023a; Yang et al., 2024; Chu et al., 2024; Ma et al., 2024). However, these NeRF-based methods still face challenges in achieving real-time rendering performance. This limitation is addressed by 3D Gaussian Splatting (Kerbl et al., 2023), which offers faster rendering while maintaining high visual quality. Unlike earlier Gaussian-based methods (Xu et al., 2024b; Wu et al., 2024; Tang et al., 2025) that rely on extensive individual-specific optimization, recent works such as GAGAvatar (Chu & Harada, 2024) and LAM (He et al., 2025) propose one-shot reconstruction frameworks for head avatars. However, these methods still face challenges such as inefficient and redundant Gaussian

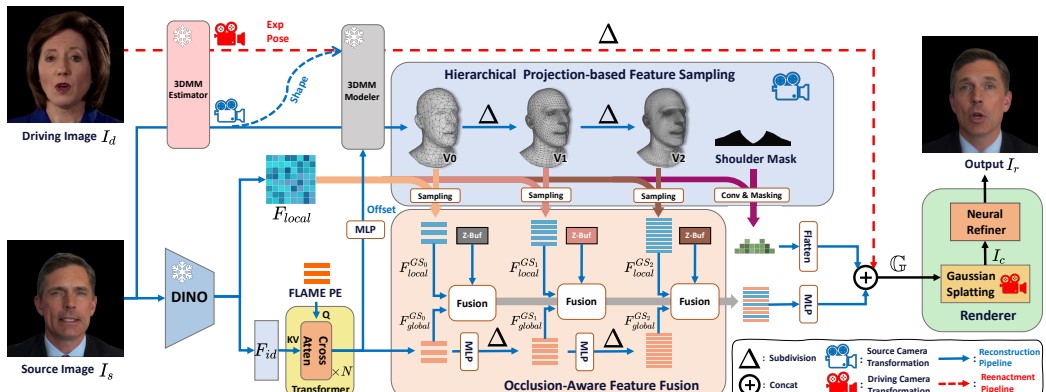

Figure 1: The overall pipeline of HIGH-Avatar framework. Our method extracts global features via cross-attention and local details via projection-based sampling, which are fused under the guidance of depth buffers. A coarse-to-fine strategy is proposed to facilitate hierarchical detail perception. The head and shoulder are predicted separately using shared features and then combined for rendering.

utilization, excessive computational demands, and incomplete head representation, which we aim to address in this paper.

**Hierarchical Gaussian Representation** The hierarchical representation has been widely applied for efficiently modeling multi-scale or structured data. HiSplat (Tang et al., 2024) introduces a hierarchical Gaussian splatting framework for sparse-view reconstruction, utilizing coarse-grained Gaussians for large-scale structures and fine-grained Gaussians for texture details. Dongye et al. (2024) integrates LOD into Gaussian avatars via hierarchical embedding, achieving a balance between visual quality and computational costs. Teotia et al. (2024) takes a hierarchical representation to capture the complex dynamics of facial expressions and head movements. In our work, we employ hierarchical representation to support efficient modeling and dynamic LOD rendering at runtime.

## 3 METHOD

Figure 1 illustrates the overall pipeline of our method. Given a source image, we first extract both local and identity features using DINOv2 (Oquab et al., 2023) and estimate the 3D head mesh via a 3DMM modeler. The Hierarchical Projection-based Feature Sampling (HPFS) module then projects the head mesh onto the image plane to sample local features at corresponding coordinates. Concurrently, the global feature is obtained through cross-attention, where FLAME positional embeddings serve as queries. Subsequently, the Occlusion-Aware Feature Fusion(OAFF) module fuses the global and local features under the guidance of depth buffers, ensuring spatial coherence and occlusion plausibility. Features from non-head regions are further integrated with head-related features to jointly predict Gaussian attributes for splatting rendering. Finally, a neural renderer generates a refined output image based on the coarse splatted feature maps. During training, we progressively subdivide the meshes along with their corresponding features, enabling the network to capture hierarchical details in a coarse-to-fine manner, which enhances both training stability and reconstruction accuracy. The details of each module are explained in the subsequent sections.

### 3.1 HIERARCHICAL GLOBAL-LOCAL FEATURE EXTRACTION

For the task of generating a 3D head model from a single image, the core objective is to establish a 2D-to-3D feature mapping mechanism that transforms image features into 3D spatial features. To incorporate statistical priors on head geometry, we employ the FLAME (Li et al., 2017) model as the 3D head representation, which comprises $N_0 = 5023$ vertices. We leverage DINOv2 (Oquab et al., 2023) to extract both local features $F_{local}$ and identity features $F_{id}$ from the source image $I_s$ following Chu & Harada (2024). For the $F_{id}$, we assign a learnable positional encoding to each vertex of FLAME as a query, and employ multiple cross-attention blocks to extract global features $F_{global}^{GS_0}$. The $F_{global}^{GS_0}$ is then used to predict vertex offsets via a MLP $\Phi_{\text{offset}}$ to improve the precision

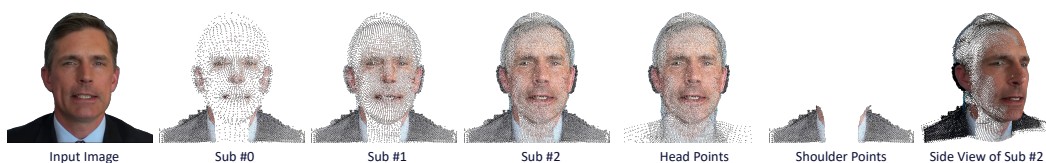

Input Image      Sub #0      Sub #1      Sub #2      Head Points      Shoulder Points      Side View of Sub #2

Figure 2: As the subdivision level increases, the resolution of head meshes and features are progressively refined. The Gaussians for the head and shoulders are predicted separately and integrated via cross-region combination. The head Gaussian counts for Sub #0, Sub #1, and Sub #2 are 5K, 20K, and 80K, respectively. See Table 5 in the appendix for more details.

of the estimated head mesh $T_p$ via 3DMM modeler:

$$T_p(\vec{\beta}, \vec{\theta}, \vec{\psi}) = \overline{T} + B_S(\vec{\beta}; S) + B_P(\vec{\theta}; P) + B_E(\vec{\psi}; E) + \Phi_{\text{offset}}(F_{global}^{GS_0}) \tag{1}$$

where $\overline{T}$ is the template mesh, and $B_S$, $B_P$ and $B_E$ represent shape, pose, and expression blend-shapes respectively. The initial head vertices $V_0$ are obtained using a standard skinning function $W$:

$$V_0 = W(T_p(\vec{\beta}, \vec{\theta}, \vec{\psi}), \boldsymbol{J}(\vec{\beta}), \vec{\theta}, \boldsymbol{\mathcal{W}}) \tag{2}$$

where $W$ rotates the morphed mesh $T_p$ around joints $\boldsymbol{J}$ and smooths it using blendweights $\boldsymbol{\mathcal{W}}$. We project $V_0$ into the image space to obtain the corresponding pixel coordinates for each vertex, and perform look-up sampling on $F_{local}$ to extract per-vertex features, denoted as $F_{local}^{GS_0}$. As noted in LAM (He et al., 2025), the original number of $V_0$ is insufficient for detailed modeling, so we introduce a coarse-to-fine strategy that progressively subdivides the mesh $V_k$ and its associated global features $F_{global}^{GS_k}$ during training:

$$F_{global}^{GS_{k+1}}, V_{k+1} = \Delta(\Phi_k(F_{global}^{GS_k}), V_k), \quad 0 \le k \le K \tag{3}$$

where $\Delta$ denotes the mesh subdivision operation, $\Phi_k$ is an MLP network, and $k$ indicates the subdivision level. With the refined vertices $V_k$, the corresponding local feature can be obtained as:

$$F_{local}^{GS_k} = Sampling(\mathrm{P}(V_k), F_{local}), \quad 0 \le k \le K \tag{4}$$

where $\mathrm{P}$ is the camera projective transformation. As $k$ increases, the resolution of head meshes and features are continuously refined. Figure 2 shows the subdivided vertices. To balance quality and efficiency, we set the maximum subdivision level to $K = 2$, resulting in 79, 936 vertices.

Notably, unlike LAM which performs costly cross-attention across all 80K vertices, our method computes cross-attention at the initial level with $N_0 = 5K$ vertices. High-resolution geometry is then incrementally refined through efficient subdivision and sampling, significantly reducing computational and memory costs while maintaining reconstruction quality.

## 3.2 OCCLUSION-AWARE FEATURE FUSION

Once hierarchical global and local features are extracted, we propose an occlusion-aware fusion strategy guided by the depth buffer to ensure robust feature integration. During the rasterization of the 3D mesh, occlusion culling is applied to invisible vertices. Consequently, local features sampled via projection are accurate for visible vertices but may be ambiguous for occluded ones, as their corresponding 2D image features are absent. On the other hand, the global features contain high-level semantic information from the input image, allowing them to infer plausible representations for occluded regions but lacking high-frequency details. Building upon this observation, we leverage the depth buffer to identify and retain only high-confidence local features from visible vertices, selectively fusing them with global features. Specifically, when each vertex is projected into the image space, its depth value is compared with the depth buffer at the corresponding pixel location. Based on this comparison, a binary visibility mask $M^{GS_k} \in \{0, 1\}^{N_k}$ is constructed. Formally, for each vertex $v_i \in V_k$, let $z_i$ be its depth in camera space, and $\hat{z}_i$ be the depth value recorded in the depth buffer. The mask $M^{GS_k} \in \{0, 1\}^{N_k}$ is defined as:

$$M_i^{GS_k} = \begin{cases} 1, & \text{if } z_i = \hat{z}_i \\ 0, & \text{if } z_i > \hat{z}_i \end{cases}, \quad \forall i = 1, 2, \ldots, N_k \tag{5}$$

Here, $M_i^{GS_k} = 1$ indicates that the vertex $v_i$ is visible, while $M_i^{GS_k} = 0$ identifies occluded vertices. The final fused head feature $F_h^{GS_k}$ is computed as:

$$F_h^{GS_k} = F_{global}^{GS_k} + F_{local}^{GS_k} \odot M^{GS_k}, \quad 0 \leq k \leq K \tag{6}$$

By effectively combining the strengths of global and local features, we achieve a more robust representation of the head vertices. A set of MLPs $\Phi$ are then employed to regress the Gaussian attributes for each head vertex, including color $c_h$, opacity $o_h$, scale $s_h$, and rotation $r_h$:

$$c_h, o_h, s_h, r_h = \Phi_{c,o,s,r}(F_h^{GS_K}) \tag{7}$$

The positions $p_h$ are directly derived from the subdivided mesh $V_K$, resulting in a full set of Gaussian parameters $\mathbb{H}$ that describe the head model:

$$\mathbb{H} = \{c_h, o_h, s_h, r_h, p_h = V_K\} \tag{8}$$

### 3.3 Multi-region Modeling and Integration

The FLAME model lacks sufficient vertex coverage in the shoulder region, resulting in coarse and blurry reconstructions in prior methods. To address this limitation, inspired by Wu et al. (2024), we perform image segmentation on the source image to obtain a shoulder mask $M_s$. The extracted local features $F_{local}$ are passed through a convolutional neural network to generate a feature plane, where separate channels encode the Gaussian rendering attributes. The shoulder-relevant region is then isolated using the mask, obtaining the corresponding parameters as follows:

$$c_s, o_s, s_s, r_s, O_s = \text{Flatten}(\text{Conv}(F_{local}^{GS}) \odot M_s) \tag{9}$$

where $O_s$ represents the offset for shoulder points. For position estimation, we generate the 3D shoulder points $\hat{p}_s$ on an image-aligned plane in world space based on the given camera transformation and feature plane resolution. These points are paired with their corresponding direction vectors $n_s$. The final shoulder points are then calculated as:

$$\mathbb{S} = \{c_s, o_s, s_s, r_s, p_s = \hat{p}_s + O_s \cdot n_s\} \tag{10}$$

Finally, we concatenate the head and shoulder parameter sets along the attribute dimensions to form a complete Gaussian parameter set $\mathbb{G}$ that covers both head and shoulder regions:

$$\mathbb{G} = \mathbb{H} \oplus \mathbb{S} \tag{11}$$

### 3.4 Reenactment and Rendering

After reconstructing the Gaussian head avatar $\mathbb{G}$, our framework enables efficient reenactment, allowing the model to mimic facial expressions and head movements observed in a target video. As shown in Figure 1, given a driving image $I_d$, the expression and pose parameters are extracted using a 3DMM estimator. These driving parameters are then combined with the identity-related parameters derived from $I_s$ to generate novel FLAME vertices. The FLAME vertices are further refined via subdivisions, resulting in the final head positions $p_h$ used for Gaussian rendering. Notably, the reenactment process only needs to update the positional component $p_h$ in the Gaussian parameter set $\mathbb{G}$ (the red dashed line in Figure 1), enabling real-time rendering of avatar animations. To enhance the expressiveness of the Gaussian representation, we adopt a rendering pipeline inspired by Chu & Harada (2024). Specifically, instead of rendering RGB values, we predict multi-channel feature maps, where the first three channels encode a coarse RGB image $I_c$. The feature maps are subsequently refined by a UNet-based neural refiner to produce the final high-quality output $I_r$.

### 3.5 Learning Strategy

The training process is conducted on a large-scale human video dataset in a self-supervised manner. For each video, we randomly sample two frames and assign one as the source image $I_s$ and the other as the driving image $I_d$. The objective is to train the network to generate an output image that closely resembles the driving image in both appearance and motion.

To achieve this, we employ a multi-component loss function, combining L2 loss, SSIM loss, and perceptual loss, applied to both the coarse and refined output images. Additionally, to constrain the

displacement of FLAME model vertices, we impose a regularization term on the vertex offsets. The total loss is defined as:

$$\mathcal{L} = \lambda_1 \mathcal{L}_2(I_d, I_c \& I_r) + \lambda_2 \mathcal{L}_{\text{SSIM}}(I_d, I_c \& I_r) + \lambda_3 \mathcal{L}_{\text{percep}}(I_d, I_c \& I_r) + \lambda_4 \mathcal{L}_{\text{reg}} \qquad (12)$$

where $\mathcal{L}(A, B\&C)$ denotes $\mathcal{L}(A, B) + \mathcal{L}(A, C)$, $\mathcal{L}_2$, $\mathcal{L}_{\text{SSIM}}$, and $\mathcal{L}_{\text{percep}}$ are computed on both the $I_c$ and refined $I_r$, and $\mathcal{L}_{\text{reg}} = \|\text{offset}\|_2$ penalizes large vertex displacements.

## 4 EXPERIMENT

### 4.1 DATASETS AND SETTINGS

**Datasets.** Our model is trained on the VFHQ (Xie et al., 2022) dataset, which contains video clips from a variety of interview scenarios. To ensure temporal diversity, we uniformly sample frames from each video following previous works (Chu & Harada, 2024; He et al., 2025), leading to a total of 766, 263 frames across 15, 204 video clips. All images are cropped to focus on the head region, resized to $512 \times 512$ pixels for consistency, and further processed with camera pose tracking, FLAME parameter estimation, and background removal as described in prior works (Chu & Harada, 2024; Chu et al., 2024). For evaluation, we adopt the official test split of the VFHQ dataset, comprising 2, 500 frames from 50 videos. The first frame of each video is used as the source image, while the remaining frames serve as driving and target images for reenactment. Additionally, we validate our method on the HDTF dataset (Zhang et al., 2021) using the standard test split introduced in Ma et al. (2023) and Li et al. (2023b), which includes 19 video sequences.

**Implementation Details.** Our transformer network for extracting global features consists of two decoder layers, each with 8 attention heads. The dimension of the FLAME positional encoding is set to 256. Rather than relying on an external semantic segmentation model, we derive the shoulder region mask by calculating the difference between the portrait mask at the bottom of the image and the depth buffer mask in a straightforward manner. The average number of Gaussian points in the shoulder region is 9K (see Table 5 in the appendix). During training, the weights of the DINOv2 and 3DMM estimator modules are frozen. We train the entire model on a single NVIDIA A100 GPU for 6 epochs using the Adam optimizer with a learning rate of $1 \times 10^{-4}$ and a batch size of 8. The subdivide levels are gradually increased based on the training stage. We set the loss parameters $\lambda_1 = 10$, $\lambda_2 = 1$, and $\lambda_3 = \lambda_4 = 0.1$. More details are provided in the appendix B.

**Evaluation Metrics.** To comprehensively evaluate the performance of both self- and cross-identity reenactment, we employ a multi-faceted assessment framework that incorporates a variety of quantitative metrics. For self-reenactment scenarios where ground-truth data is available, we assess the quality of generated images using three widely adopted objective measures: Peak Signal-to-Noise Ratio (PSNR), Structural Similarity Index (SSIM), and Learned Perceptual Image Patch Similarity (LPIPS) (Zhang et al., 2018). These metrics provide reliable comparisons between synthesized outputs and reference ground-truth images. To evaluate identity preservation, we compute the cosine distance between face recognition features extracted from the source and reenacted images, following the methodology proposed in Deng et al. (2019a). For assessing the accuracy of expression and pose transfer, we utilize a 3D Morphable Model (3DMM) estimator (Deng et al., 2019b) to calculate the Average Expression Distance (AED) and Average Pose Distance (APD). Additionally, we use facial landmark detection (Bulat & Tzimiropoulos, 2017) to measure the Average Keypoint Distance (AKD), which provides further insight into the precision of motion control during animation. In the case of cross-identity reenactment, where ground-truth data is not available, we adopt an evaluation protocol based on Consistency of Identity Similarity (CSIM), AED, and APD—metrics aligned with those used in recent studies (Chu & Harada, 2024; He et al., 2025). This ensures comparability across different methods and enables meaningful analysis.

### 4.2 BASELINE METHODS

We conduct a comprehensive comparison between our method and state-of-the-art 3D avatar approaches, including ROME (Khakhulin et al., 2022), StyleHeat (Yin et al., 2022), OTAvatar (Ma et al., 2023), HideNeRF (Li et al., 2023a), GOHA (Li et al., 2023b), CVTHead (Ma et al., 2024), GPAvatar (Chu et al., 2024), Real3DPortrait (Ye et al., 2024), Portrait4D (Deng et al., 2024a), Portrait4D-v2 (Deng et al., 2024b), GAGAvatar (Chu & Harada, 2024), and LAM (He et al., 2025). For each baseline, we utilize its official implementation to produce results.

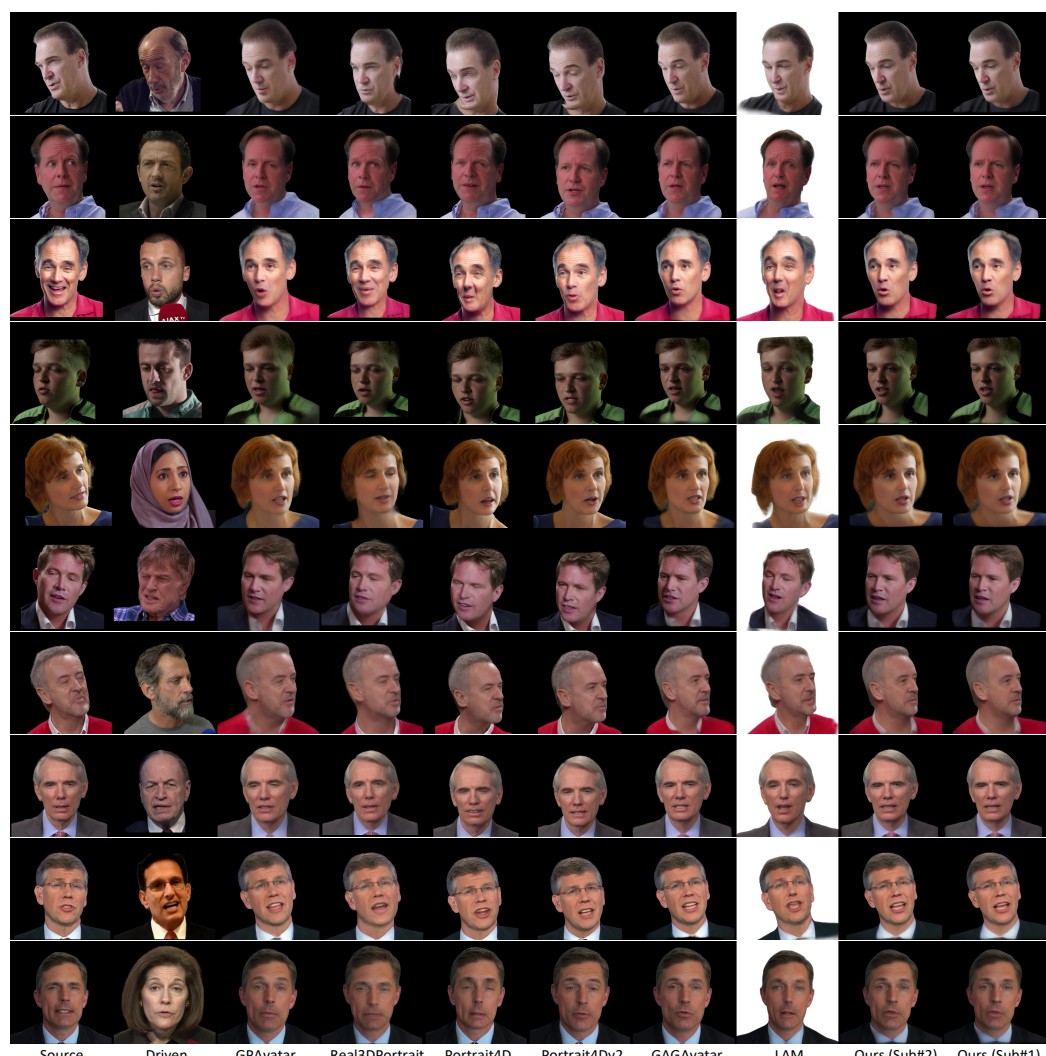

Figure 3: Cross-identity reenactment results on VFHQ and HDTF datasets.

### 4.3 QUALITATIVE EVALUATION

We compare our method with baseline approaches on the VFHQ and HDTF datasets, the visual results are presented in Figure 3. Our method demonstrates superior performance over existing approaches in terms of reconstruction detail, identity preservation, and reenactment consistency. Compared with recently Gaussian-based works, our approach outperforms He et al. (2025) in capturing facial details (*e.g.*, the mouth) and expression-dependent dynamic textures (*e.g.*, forehead wrinkles) thanks to the hierarchical presentation and neural rendering strategy. Additionally, we achieve higher reconstruction quality than Chu & Harada (2024) while using significantly fewer Gaussian points. In contrast to the unrealistic tilt and blur in the shoulder region observed in He et al. (2025) and Chu & Harada (2024), our multi-region modeling strategy effectively improves the quality of the shoulder area. Moreover, our low-resolution results (Sub #1 with ∼29K Gaussian points) shown in the last column of Figure 3 maintain comparable visual quality, making them well-suited for deployment in high-speed applications or on hardware with limited computational resources.

### 4.4 QUANTITATIVE EVALUATION

We report the quantitative results in Table 1 and Table 2. Our method (Sub #2) outperforms existing approaches across all reconstruction metrics (PSNR, SSIM, and LPIPS), as well as identity, ex-

Table 1: **Quantitative results on the VFHQ dataset.** The first, second, and third best-performing methods are highlighted. The Sub # indicates the subdivision level for inference.

| Method | Self Reenactment | | | | | | | Cross Reenactment | | |
|---|---|---|---|---|---|---|---|---|---|---|
| | PSNR↑ | SSIM↑ | LPIPS↓ | CSIM↑ | AED↓ | APD↓ | AKD↓ | CSIM↑ | AED↓ | APD↓ |
| StyleHeat | 19.95 | 0.726 | 0.211 | 0.537 | 0.199 | 0.385 | 7.659 | 0.407 | 0.279 | 0.551 |
| ROME | 19.96 | 0.786 | 0.192 | 0.701 | 0.138 | 0.186 | 4.986 | 0.530 | 0.259 | 0.277 |
| OTAvatar | 17.65 | 0.563 | 0.294 | 0.465 | 0.234 | 0.545 | 18.19 | 0.364 | 0.324 | 0.678 |
| HideNeRF | 19.79 | 0.768 | 0.180 | 0.787 | 0.143 | 0.361 | 7.254 | 0.514 | 0.277 | 0.527 |
| GOHA | 20.15 | 0.770 | 0.149 | 0.664 | 0.176 | 0.173 | 6.272 | 0.518 | 0.274 | 0.261 |
| CVTHead | 18.43 | 0.706 | 0.317 | 0.504 | 0.186 | 0.224 | 5.678 | 0.374 | 0.261 | 0.311 |
| GPAvatar | 21.04 | 0.807 | 0.150 | 0.772 | 0.132 | 0.189 | 4.226 | 0.564 | 0.255 | 0.328 |
| Real3DPortrait | 20.88 | 0.780 | 0.154 | 0.801 | 0.150 | 0.268 | 5.971 | 0.663 | 0.296 | 0.411 |
| Portrait4D | 20.35 | 0.741 | 0.191 | 0.765 | 0.144 | 0.205 | 4.854 | 0.596 | 0.286 | 0.258 |
| Portrait4D-v2 | 21.34 | 0.791 | 0.144 | 0.803 | 0.117 | 0.187 | 3.749 | 0.656 | 0.286 | 0.273 |
| GAGAvatar | 21.83 | 0.818 | 0.122 | 0.816 | 0.111 | 0.135 | 3.349 | 0.633 | 0.253 | 0.247 |
| LAM | 22.65 | 0.829 | 0.109 | 0.822 | 0.102 | 0.134 | 2.059 | 0.651 | 0.250 | 0.356 |
| Ours (Sub #2) | 22.72 | 0.831 | 0.091 | 0.869 | 0.088 | 0.111 | 2.045 | 0.660 | 0.235 | 0.257 |
| Ours (Sub #1) | 22.68 | 0.830 | 0.094 | 0.858 | 0.089 | 0.112 | 2.055 | 0.644 | 0.233 | 0.260 |
| Ours (Sub #0) | 22.18 | 0.817 | 0.102 | 0.855 | 0.134 | 0.142 | 2.790 | 0.616 | 0.254 | 0.279 |

Table 2: **Quantitative results on the HDTF dataset.**

| Method | Self Reenactment | | | | | | | Cross Reenactment | | |
|---|---|---|---|---|---|---|---|---|---|---|
| | PSNR↑ | SSIM↑ | LPIPS↓ | CSIM↑ | AED↓ | APD↓ | AKD↓ | CSIM↑ | AED↓ | APD↓ |
| StyleHeat | 21.41 | 0.785 | 0.155 | 0.657 | 0.158 | 0.162 | 4.585 | 0.632 | 0.271 | 0.239 |
| ROME | 20.51 | 0.803 | 0.145 | 0.738 | 0.133 | 0.123 | 4.763 | 0.726 | 0.268 | 0.191 |
| OTAvatar | 20.52 | 0.696 | 0.166 | 0.662 | 0.180 | 0.170 | 8.295 | 0.643 | 0.292 | 0.222 |
| HideNeRF | 21.08 | 0.811 | 0.117 | 0.858 | 0.120 | 0.247 | 5.837 | 0.843 | 0.276 | 0.288 |
| GOHA | 21.31 | 0.807 | 0.113 | 0.725 | 0.162 | 0.117 | 6.332 | 0.735 | 0.277 | 0.136 |
| CVTHead | 20.08 | 0.762 | 0.179 | 0.608 | 0.169 | 0.138 | 4.585 | 0.591 | 0.242 | 0.203 |
| GPAvatar | 23.06 | 0.855 | 0.104 | 0.855 | 0.114 | 0.135 | 3.293 | 0.842 | 0.268 | 0.219 |
| Real3DPortrait | 22.82 | 0.835 | 0.103 | 0.851 | 0.138 | 0.137 | 4.640 | 0.903 | 0.299 | 0.238 |
| Portrait4D | 20.81 | 0.786 | 0.137 | 0.810 | 0.134 | 0.131 | 4.151 | 0.793 | 0.291 | 0.240 |
| Portrait4D-v2 | 22.87 | 0.860 | 0.105 | 0.860 | 0.111 | 0.111 | 3.292 | 0.857 | 0.262 | 0.183 |
| GAGAvatar | 23.13 | 0.863 | 0.103 | 0.862 | 0.110 | 0.111 | 2.985 | 0.851 | 0.231 | 0.181 |
| LAM | 23.43 | 0.873 | 0.097 | 0.865 | 0.101 | 0.093 | 1.965 | 0.849 | 0.230 | 0.229 |
| Ours (Sub #2) | 24.14 | 0.875 | 0.061 | 0.943 | 0.080 | 0.064 | 1.806 | 0.886 | 0.226 | 0.155 |
| Ours (Sub #1) | 24.06 | 0.874 | 0.063 | 0.937 | 0.081 | 0.066 | 1.834 | 0.881 | 0.227 | 0.155 |
| Ours (Sub #0) | 23.85 | 0.868 | 0.067 | 0.942 | 0.121 | 0.085 | 2.381 | 0.872 | 0.246 | 0.156 |

pression, and pose consistency. Remarkably, our low-resolution LOD Sub #1 surpasses LAM (80K Gaussians) and GAGAvatar (180K Gaussians) on both datasets using only 29K Gaussian points, demonstrating the effectiveness of our hierarchical feature extraction and fusion strategy. Further information can be found in Figure 8 of the appendix.

We further report the inference efficiency in Table 3. Our method achieves an inference speed of 85 FPS on an A100 GPU and 126 FPS on the consumer-grade RTX 4090 GPU, using the native PyTorch framework and the official implementation of 3D Gaussian Splatting. Compared to existing neural-rendering-based methods, our approach attains the highest inference speed. Moreover, our method outperforms LAM (280 FPS on A100 GPU without neural rendering) in terms of geometric details and dynamic textures, achieving an optimal balance between efficiency and visual quality.

## 4.5 ABLATION STUDIES

**Subdivision Times.** To evaluate the effect of subdivision levels, we compare the results of our model with varying subdivision levels. The visual results are presented in Figure 5. Our observations indicate that higher subdivision levels capture more high-frequency details, such as hair and wrinkles, leading to improved reconstruction quality. Quantitative comparisons are provided in Table 1 and Table 2, which further validate this trend. Additionally, increasing the number of subdivisions enhances the reconstruction quality at the cost of reduced inference speed, as shown in

Table 3: The reenactment speed of neural-rendering-based methods measured in FPS, averaged over 100 frames. Driving parameters estimation time is excluded as they can be precomputed.

| Methods | **A100 GPU** | | | | | | | | **RTX 4090 GPU** | | |
| | StyleHeat | ROME | HideNeRF | CVTHead | Real3D | P4D-v2 | GAGavatar | Ours (Sub #2) | Sub #2 | Sub #1 | Sub #0 |
|---|---|---|---|---|---|---|---|---|---|---|---|
| **FPS** | 19.82 | 11.21 | 9.73 | 18.09 | 4.55 | 9.62 | 67.12 | **85.94** | 126.44 | 148.04 | 152.57 |

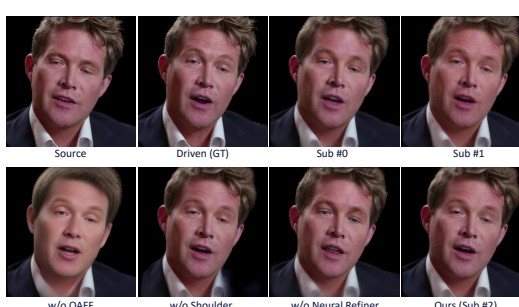

Source    Driven (GT)    Sub #0    Sub #1

w/o OAFF    w/o Shoulder    w/o Neural Refiner    Ours (Sub #2)

Table 4: Ablation results on the VFHQ dataset.

| Methods | PSNR↑ | SSIM↑ | LPIPS↓ | CSIM↑ |
|---|---|---|---|---|
| w/o OAFF | 21.21 | 0.802 | 0.128 | 0.429 |
| w/o Refiner | 21.42 | 0.809 | 0.115 | 0.842 |
| w/o Shoulder | 22.42 | 0.828 | 0.099 | 0.867 |
| Ours | **22.72** | **0.831** | **0.091** | **0.869** |

Figure 5: Our local-global feature fusion (OAFF) and multi-region fusion strategy significantly improve identity consistency and completeness in non-head regions. The neural refiner further boosts visual fidelity, especially for dynamic facial expressions.

Table 3. Notably, even our low-resolution LOD demonstrates competitive performance compared to existing methods, highlighting the effectiveness of our framework.

**Local-Global and Multi-Region Feature Fusion.** We conduct an ablation study by removing the global-local feature fusion in the OAFF module and using only global features. As shown in Figure 5 and Table 4, the absence of sampled local features significantly impacts identity consistency. Furthermore, when the shoulder region is excluded during rendering, the results display an incomplete and blurry appearance of the shoulder, as illustrated in Figure 5.

**Neural Rendering.** We evaluate the effectiveness of the neural refiner module. As shown in Table 4, the neural refiner contributes to improvements in both visual fidelity and identity consistency. The last two images in Figure 5 further demonstrate that the neural refiner enhances fine details such as teeth and plays a key role in capturing expression-dependent features, including forehead wrinkles during eyebrow raising.

## 5 CONCLUSION

In this paper, we present a novel hierarchical framework for Gaussian head avatar reconstruction in a feed-forward manner. Our method enables dynamic level-of-detail (LOD) rendering at runtime, offering flexibility to accommodate varying device capabilities and inference speed requirements. Our model exhibits superior reconstruction and reenactment performance with significantly reduced computational cost. This is achieved by an efficient multi-level global and local feature extraction and a coarse-to-fine refinement strategy, as well as an occlusion-aware fusion mechanism. Moreover, our multi-region modeling scheme effectively enhances the visual fidelity of shoulder areas. Extensive experiments on two public datasets demonstrate that our approach outperforms state-of-the-art methods in terms of reconstruction quality, reenactment performance, and computational efficiency.

**Limitations and Future Work.** Despite achieving strong results, our approach has two main limitations. First, the 3D Gaussian head model relies on the FLAME prior and accurate 3D morphable model (3DMM) tracking. However, FLAME does not capture fine facial dynamics—such as tongue motion, hair deformation, and subtle expressions—limiting the expressiveness of the generated avatars. Second, training solely on monocular videos reduces robustness to large viewpoint changes, leading to appearance inconsistencies and identity drift when the driving and source views differ significantly. To address these issues, we plan to incorporate multi-view datasets for training, which will enhance spatial understanding and improve robustness across varying viewpoints.

## 6    REPRODUCIBILITY STATEMENT

We provide the code in the supplementary material. Our implementation partially builds upon FLAME and GAGAvatar, and we sincerely appreciate the authors for sharing their valuable resources. To ensure anonymity during submission, we have temporarily removed the header comments from certain code files, which will be included in the final public release. In the appendix B, we provide further details on the model implementation and data preprocessing pipeline.

## 7    ETHICAL DISCUSSION

Our method enables high-fidelity, animatable 3D head avatar generation with potential applications in video production, digital communication, and other domains. However, like other advanced generative models, it could be misused to create deceptive or non-consensual synthetic content (commonly known as "deepfakes") that may mislead, manipulate, or infringe on personal privacy. We firmly oppose such misuse and emphasize that our work is intended solely for legitimate, consent-based applications. To mitigate potential risks, we propose the following safeguards:

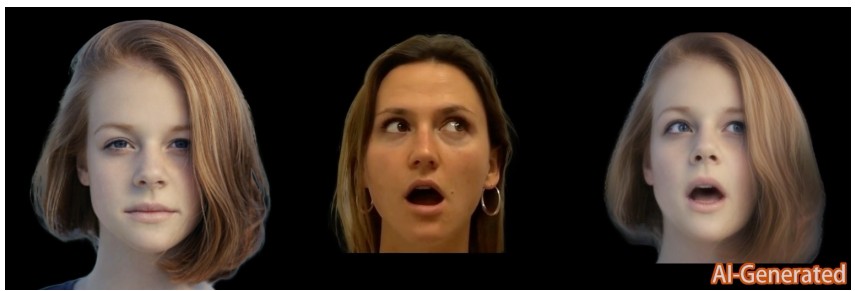

Figure 6: Visible watermarks will be embedded in all generated images and videos, clearly indicating that the content is AI-generated.

- **Visible and invisible watermarking.** We integrate visible watermarking mechanisms into our released code, as shown in Figure 6. Visible watermarks will be embedded in all generated images and videos, clearly indicating that the content is AI-generated, enabling viewers to easily distinguish synthetic media from authentic recordings. In addition, we plan to adopt robust invisible watermarking techniques (Tancik et al., 2020) that embed and reliably decode arbitrary data (*e.g.*, hyperlinks) in a perceptually invisible manner, while remaining resilient to real-world distortions such as compression, printing, and re-photography. These watermarks are designed to be difficult to remove without degrading visual quality.

- **Strict licensing.** Our code and models will be released under a restrictive license that prohibits the creation of avatars based on real individuals without explicit consent, particularly for commercial purposes. The license further restricts usage to ethical and non-deceptive applications, and any violation can be traced through the embedded watermarking system.

To summarize, while our method advances the state of animatable head generation, we acknowledge its dual-use potential. Through technical measures like watermarking and policy-level controls via licensing, we aim to minimize the risk of abuse. As technology developers, we have the responsibility to build safeguards into our systems. However, preventing misuse requires broader efforts, from platform policies, legal frameworks, and user awareness. We call on researchers, developers, and content creators to exercise ethical judgment and social responsibility when deploying generative avatar systems. With appropriate oversight and responsible use, our work can contribute positively to immersive communication and other beneficial applications.

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

## A  STATEMENT ON THE USE OF LARGE LANGUAGE MODELS

Throughout the paper writing process, we leveraged advanced language models such as GPT-5 as auxiliary tools to assist with language refinement, grammar correction, and stylistic improvement. These tools were primarily used to enhance the clarity, fluency, and readability of the written content. However, it is important to emphasize that all research ideas, methodologies, experimental designs, and analyses presented in this work were independently conceived, developed, and executed by the research team. The use of AI-assisted tools was strictly limited to post-writing refinement and did not influence the originality or intellectual contribution of the study.

## B  REPRODUCIBILITY DETAILS

### B.1  DATASET PROCESSING

We construct our training and testing datasets by uniformly sampling frames from the videos in the VFHQ dataset. Specifically, we use 15, 204 video clips for training and 50 clips for testing. For the training set, we sample a number of frames $N$ per clip based on the video length, following the strategy proposed in Chu & Harada (2024):

- $N = 25$ if the video length is less than 200 frames,
- $N = 50$ if the video length is between 200 and 300 frames,
- $N = 75$ if the video length exceeds 300 frames.

This results in a total of 766, 263 training frames. For testing, we sample $N = 50$ frames per video, yielding 2, 500 test frames in total.

To evaluate the generalization ability of our model, we directly test it on the HDTF dataset using weights trained on the VFHQ dataset. We follow the dataset split setting from Ma et al. (2023), and uniformly sample 100 frames per video, resulting in a total of 1, 900 frames for evaluation.

### B.2  MORE IMPLEMENTATION DETAILS

We utilize a frozen DINOv2 model to extract both local and identity features from an input image.The local feature has a size of $256 \times 296 \times 296$, while the identity feature is of size $1369 \times 768$. The original FLAME mesh contains 5, 023 vertices, and its positional encoding has a size of 5, $023 \times 256$. After passing through a two-layer Transformer, we obtain a global feature map with dimensions 5, $023 \times 256$. Subdividing the global feature results in hierarchical representations at level 1 and level 2, with sizes of 20, $018 \times 256$ and 79, $936 \times 256$, respectively. The first dimension of these features corresponds to the number of vertices at each subdivision level. For the shoulder regions, we compute its mask by taking the difference between the bottom quarter of the portrait mask and the depth buffer mask, as shown in Figure 7.

During training, the number of subdivisions is progressively increased according to the training progress. Specifically, no subdivision is applied in the early phase ($\leq 10\%$ of total iterations), one level is used in the intermediate phase (10%–30%), and a random strategy with a bias towards 2 (70% for 2, 20% for 1, and 10% for 0) in the later stage. The model is trained for 6 epochs on a single NVIDIA A100 GPU using the Adam optimizer and a linear learning rate decay schedule, with an initial learning rate of $1 \times 10^{-4}$ and a batch size of 8.

## C  MESH SUBDIVISION

We apply the Loop subdivision algorithm (LOOP, 1987) to perform mesh subdivision on the FLAME model, utilizing its implementation in PyTorch3D. The Loop subdivision algorithm refines a triangle mesh by introducing a new vertex at the midpoint of each edge and dividing each triangular face into four smaller triangles. Additionally, vertex attribute vectors (*e.g.*, $F_{global}^{GS}$ in Equation 3) are subdivided by averaging the attribute values of the two vertices that form each edge.

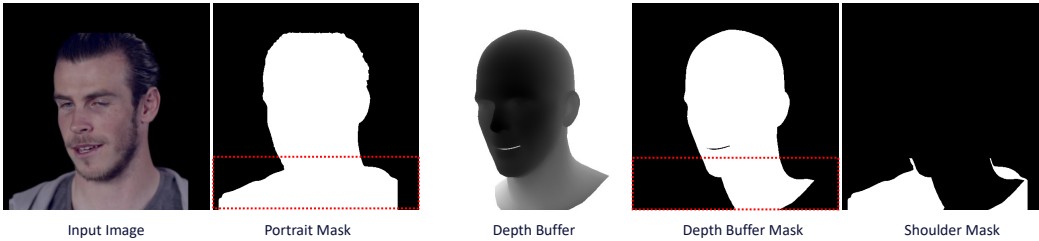

| Input Image | Portrait Mask | Depth Buffer | Depth Buffer Mask | Shoulder Mask |

Figure 7: Instead of relying on an external semantic segmentation model, we derive the shoulder region mask by calculating the difference between the portrait mask at the bottom of the image and the depth buffer mask.

In Table 5, we present the number of Gaussian points for different subdivision levels, and the shoulder region approximately adds 9K additional points. Moreover, as shown in Figure 8, our method outperforms LAM-80K and GAGAvatar (180K) using only 29K Gaussian points.

| Subdivision Times | Head Only | Average Points on VFHQ Dataset | Average Points on HDTF Dataset |
|---|---|---|---|
| #0 | 5, 023 | 13, 883 | 14, 466 |
| #1 | 20, 018 | 28, 878 | 29, 461 |
| #2 | 79, 936 | 88, 796 | 89, 379 |
| **Shoulder Points** | | +8, 860 | +9, 443 |

Table 5: The number of Gaussian points for different subdivision level. Integrating the shoulder region leads to an average increase of 8, 860 Gaussian points on the VFHQ dataset and 9, 443 on the HDTF dataset.

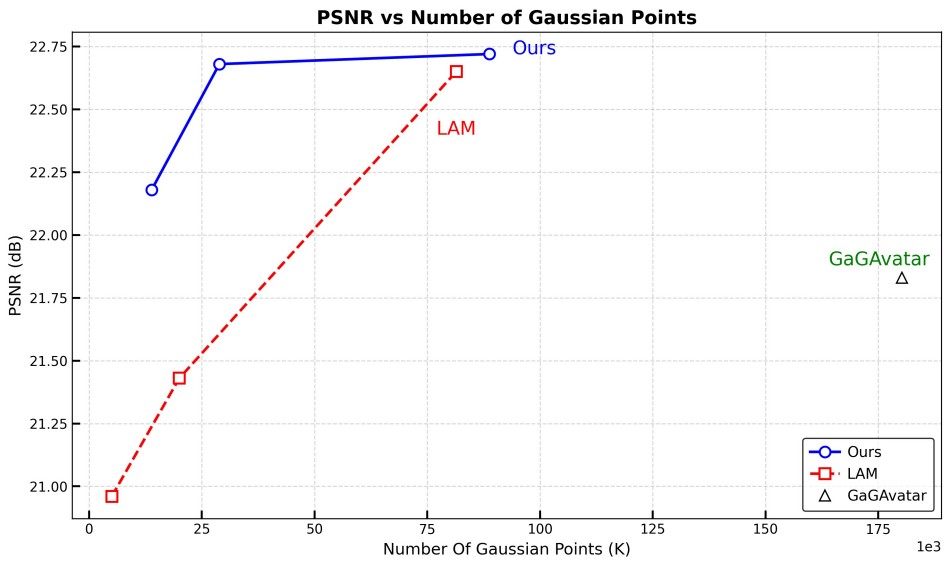

Figure 8: The correlation between Gaussian count and reconstruction performance on the VFHQ dataset. Our method achieves performance of 22.68 dB using 29K Gaussian points, surpassing LAM-80K (22.65 dB with 80K Gaussians) and GAGAvatar (21.83 dB with 180K Gaussians).

## D    ABOUT BASELINE METHODS

The quantitative performance reported in Table 1 and Table 2 is taken from the respective papers, particularly Chu & Harada (2024) and He et al. (2025). The qualitative results shown in Figure 3 are generated by running their publicly released models and code. For consistency, we set the background color to black when rendering Gaussian points for all methods following the setup in Chu & Harada (2024), except for LAM (He et al., 2025). We found that their results exhibit noticeable white contours against a black background, as illustrated in Figure 9. To preserve visual quality and maintain fidelity to the original presentation, we retained the white background as used in their work. Since LAM only released their LAM-20K model, we additionally present our results at subdivision level 1 in Figure 3 and Figure 10 for fair comparisons.

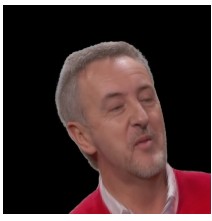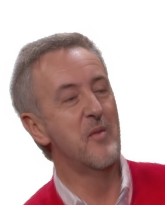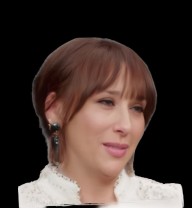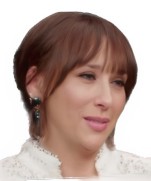

Figure 9: Results from LAM (He et al., 2025) exhibit noticeable white contours against a black background. To preserve visual quality and maintain consistency with the original presentation, we display their results using a white background.

## E    COMPUTATIONAL COST ANALYSIS

Similar to our approach, the LAM method also utilizes mesh subdivision to increase the number of points, thereby improving the Gaussian's capability to capture fine-grained details. Both methods set the number of subdivision iterations to 2. However, the computational complexity differs significantly. Let $k$ denote the subdivision level. After $k$ iterations, the number of vertices in the mesh is approximately $4^k V_0$, where $V_0$ represents the number of vertices in the original FLAME mesh. Consequently, the computational complexity of the most expensive module, the cross attention, is:

$$O(l \cdot h \cdot 4^k \cdot V_0 \cdot N_{\text{DINO}} \cdot d_{\text{head}}) \tag{13}$$

where $l$ denotes the number of transformer layers, $h$ is the number of attention heads, $N_{\text{DINO}}$ is the number of DINO features, and $d_{\text{head}}$ is the feature dimension. In our method, cross-attention is computed at the 0-th level (before any subdivision), while LAM performs it on the finest-level subdivided mesh, where the computational complexity grows exponentially with the number of subdivisions $k$. As a result, the reconstruction cost of our method is only $1/640$ that of LAM. As shown in Table 6, our training GPU-hours are more than 90% lower than LAM's.

|  | # Transformer Layers | # Attention Heads Per Layer | Attention Map Size | Feature Dimension | Training GPU Hours |
|---|---|---|---|---|---|
| **LAM** | 10 | 16 | $80\text{k} \times N_{\text{DINO}}$ | 1024 | $\sim 2600\,\text{h}$ (2 weeks $\times$ 8 GPUs) |
| **Ours** | 2 | 8 | $5\text{k} \times N_{\text{DINO}}$ | 256 | $\sim 200\,\text{h}$ |

Table 6: Comparison of training configurations for LAM and our High-Avatar.

## F    MORE QUALITATIVE RESULTS

We present additional cross-reenactment results in Figure 10 and self-reenactment results in Figure 11. Furthermore, more results on in-the-wild images are provided in Figure 12.

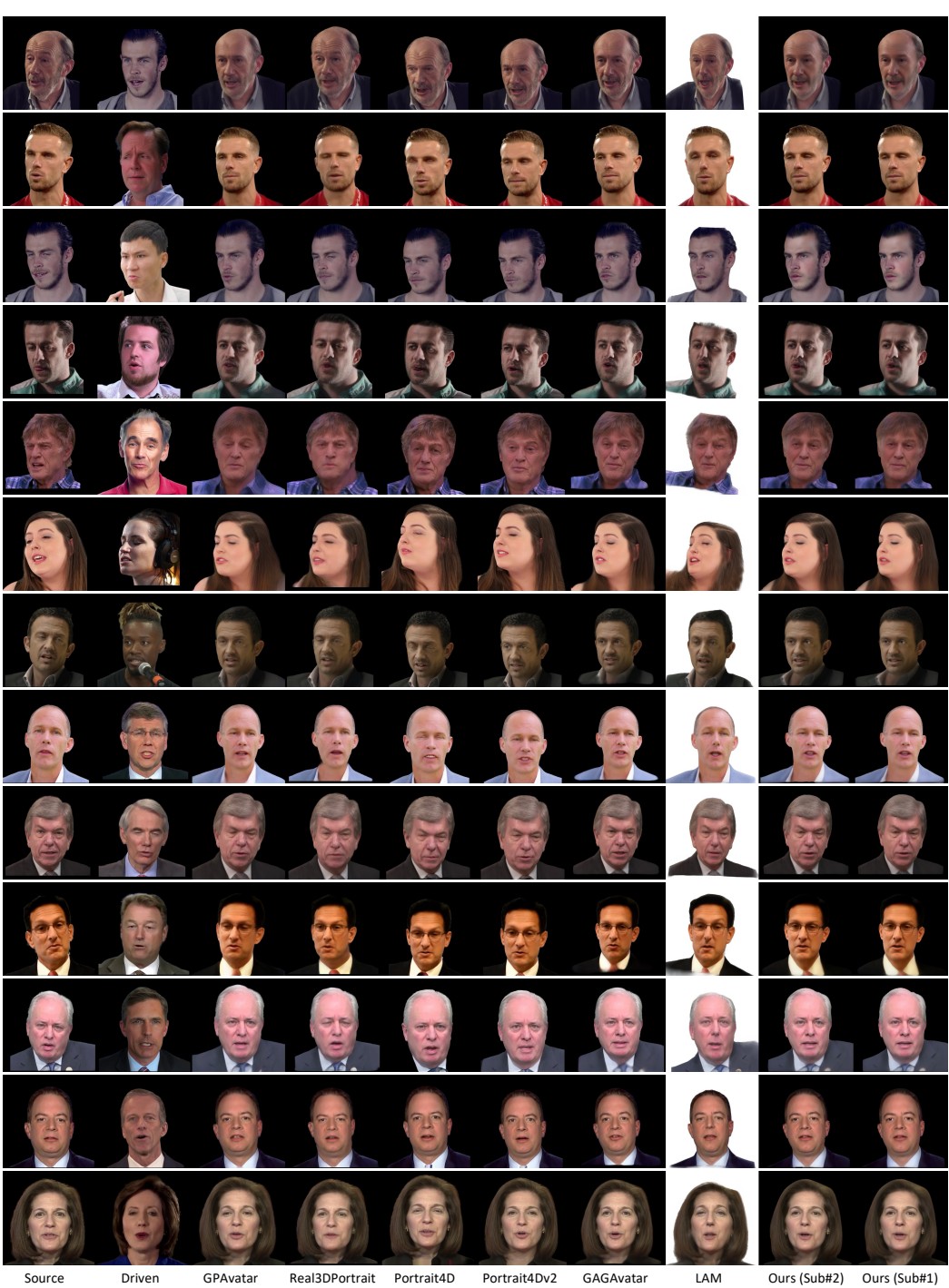

Figure 10: Cross-identity reenactment results on VFHQ and HDTF datasets.

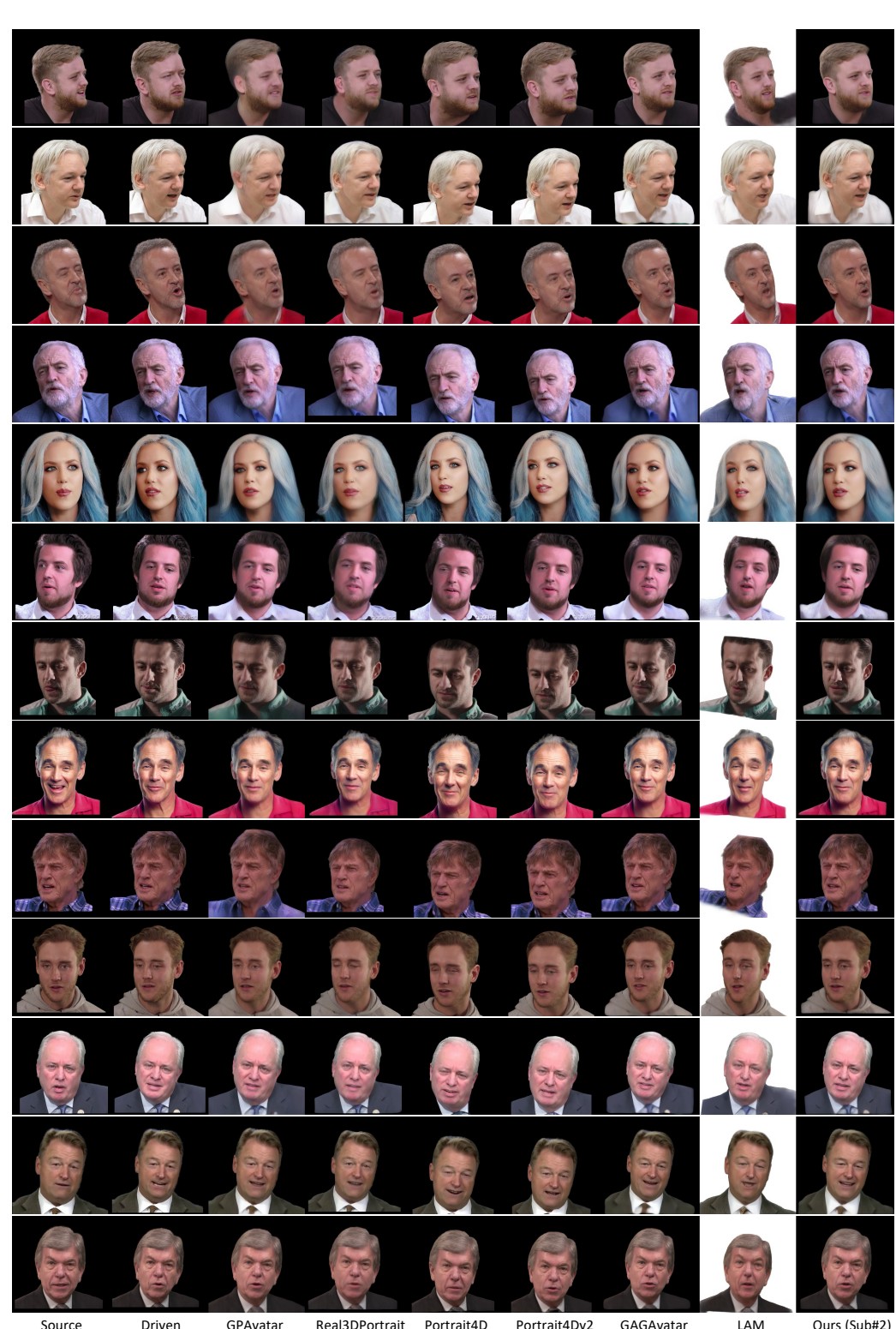

Figure 11: Self-identity reenactment results on VFHQ and HDTF datasets.

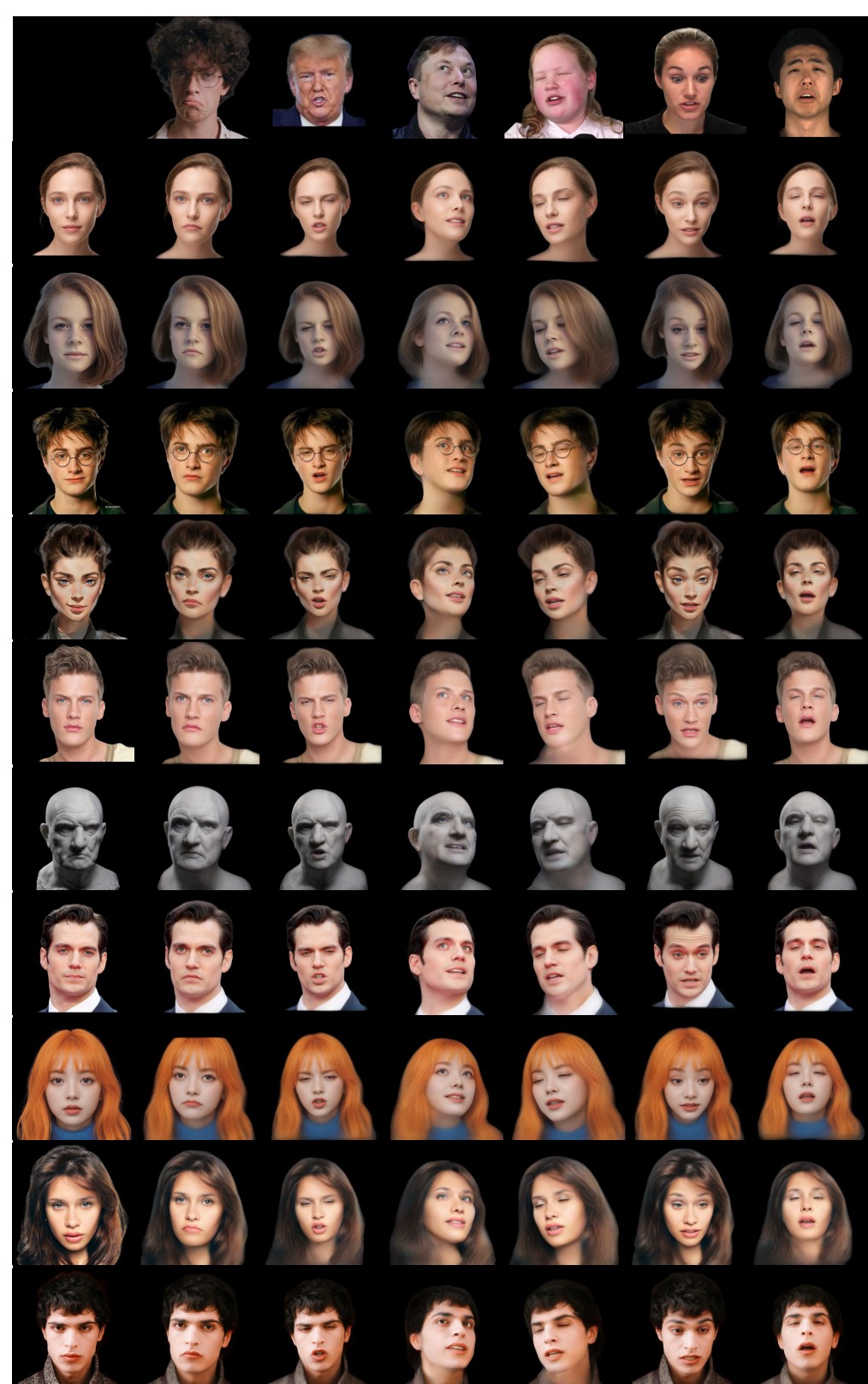

Figure 12: Cross-identity reenactment results on in-the-wild images.

