# OpenReview forum: "HIGH-AVATAR: Hierarchical Representation for One-shot Gaussian Head Avatar"
_ICLR.cc/2026/Conference — ICLR 2026 Conference Withdrawn Submission_

### Official Review · Reviewer_bp1x · 2025-10-27

**Soundness:** 3
**Presentation:** 2
**Contribution:** 2
**Rating:** 2
**Confidence:** 4

**Summary:**

This paper presents a one-shot drivable Gaussian head generation framework that leverages a multi-LOD Gaussian subdivision scheme and a depth-based feature fusion strategy to improve geometric detail and visual expressiveness. While the idea is interesting, the technical novelty and experimental validation are not convincing. Although the authors claim to outperform recent state-of-the-art methods, the supplementary videos indicate that the approach still suffers from significant visual and structural issues.

**Strengths:**

1. The paper is clearly written and the overall pipeline is easy to follow.

2. The use of a z-buffer–based feature fusion mechanism is an interesting idea that may inspire future extensions.

3. The proposed subdivision strategy can effectively reduce the computational overhead of cross-attention.

**Weaknesses:**

1. The claimed architectural novelty appears overstated. The proposed shoulder mask introduces unnecessary design complexity and resembles over-engineering. Such techniques have already been extensively explored in NeRF-based systems without offering substantial improvements. Moreover, 3D Gaussian frameworks inherently support learning positional offsets to refine geometry. The hierarchical coarse-to-fine feature sampling strategy is also insufficiently justified.
2. The experimental validation raises serious concerns. Despite the claim of outperforming SOTA methods, the supplementary videos reveal evident artifacts — the teeth are blurry, and the shoulder and boundary regions contain visible distortions. This contradicts the paper’s claimed performance advantages.
3. The reconstructed 3D head geometry shows noticeable inward deformation, especially in the hair region, as visualized in Fig. 2.
4. The choice of reference images could be improved by including more diverse facial expressions. The current demonstrations all use closed-mouth expressions, which may explain the degraded quality of the mouth and teeth regions.
5. When AIGC-generated source images are used, many fine-grained motion details are lost, indicating limited generalization capability of the method.

**Questions:**

1. The model is trained only on the VFHQ dataset. Could the authors clarify whether the model generalizes well to other datasets or in-the-wild scenarios?

2. Could the authors explain the rationale behind using the coarse-to-fine (multi-LOD) design in the architecture? Why would directly using the same number of V2 points for cross-attention be suboptimal?

---

### Official Review · Reviewer_mGeX · 2025-10-29

**Soundness:** 3
**Presentation:** 3
**Contribution:** 3
**Rating:** 4
**Confidence:** 4

**Summary:**

This paper proposes a one-shot framework for animatable 3D head avatar reconstruction using a hierarchical Gaussian representation. The model enables dynamic LOD rendering from a single image by combining transformer-based global features, projection-sampled local features, and occlusion-aware fusion guided by depth buffers. Additionally, a multi-region decomposition models head and shoulders separately, enhancing completeness.

**Strengths:**

- Novel hierarchical representation that allows scalable LOD-based rendering.

- Occlusion-aware feature fusion effectively balances local detail and global semantics.

- Multi-region modeling significantly improves shoulder reconstruction quality.

- The paper demonstrates high inference speed (85–120 FPS) while maintaining competitive visual fidelity.

- Clear ethical considerations and reproducibility statement.

**Weaknesses:**

-  Overall results appear slightly over-smoothed. Fine-grained wrinkles (e.g., frown lines, crow’s feet during squinting) are often missing, leading to a soft appearance.

-  The oral area exhibits noticeable artifacts and blur, especially in wide-open mouth expressions. This affects perceived realism.

- Although the paper claims to produce a “3D avatar,” no explicit novel-view rendering or side-view synthesis results are shown. This weakens the 3D avatar claim, even if acknowledged in the limitations.

- Comparison gaps and weaker performance vs. 2D baselines
   - Despite the claimed advantages of 3D Gaussian representations under large pose variations, the visual quality still lags significantly behind strong 2D reenactment methods such as XNEMO, even in those large-pose scenarios where the authors argue 3DGS should excel.
- the paper lacks discussion with other advanced 3D or Gaussian-based approaches (e.g., AVAT3R, HeadGaP), which would provide a more balanced and convincing evaluation of the method’s position within the current landscape.

- The proposed method is conceptually close to GAGAvatar.

**Questions:**

- How does the model handle unseen poses (e.g., ±90° head rotation) or extreme lighting?

---

### Official Review · Reviewer_Ep9Q · 2025-10-31

**Soundness:** 3
**Presentation:** 3
**Contribution:** 3
**Rating:** 6
**Confidence:** 4

**Summary:**

This paper presents HIGH-Avatar, a novel method for quickly generating high-quality 3D head avatars from a single image. By using hierarchical Gaussian representation, occlusion-aware feature fusion, and separate modeling of the head and shoulders, it achieves higher rendering quality with lower computational cost and supports real-time animation. Experiments show that it outperforms existing methods in image quality, rendering speed, and efficiency.

**Strengths:**

1. The paper is clearly written, well-structured, and rich in figures and reproducible details, making the method easy to follow and re-implement.

2. Extensive experiments on two large datasets with 12 baselines, thorough ablation studies, and multi-LOD evaluations demonstrate solid and convincing workload.

3. Novel contributions include hierarchical 3D Gaussian LOD representation, occlusion-aware global-local feature fusion, head-shoulder decomposition, and coarse-to-fine training, all working together for high-fidelity avatar generation.

4. Competitive overall performance.

**Weaknesses:**

1. Shoulder modeling relies on image-plane unfolding without 3D geometric priors, leading to texture stretching and geometric inconsistencies under large viewpoint or motion changes.

2. The method has not been tested on low-quality inputs (blur, low resolution, lighting variation, occlusion), so its robustness in real-world conditions is uncertain.

**Questions:**

1. Regarding the propagation of local-feature sampling errors:As the mesh is subdivided, local features are obtained by projecting ever-higher-resolution vertices back to the single input image. Could the authors share any insight into how projection inaccuracies (e.g., small calibration or geometry errors) are prevented from being amplified at finer levels, and whether they might shift high-frequency details such as wrinkles or beard textures?

2. On the robustness of the occlusion test:The visibility mask is derived from a standard depth buffer. Have you observed cases where hair, glasses, or other thin structures produce depth conflicts, and might this lead to visible artefacts if local features are sampled from incorrectly "visible" pixels?

3. Saturation point of Gaussian count vs. quality:Sub#2 model already achieves excellent results with ≈29 k Gaussians. I’m curious whether you have experimented with even denser sets (say 100 k–200 k). At what point do further additions stop improving PSNR/LPIPS, and did you plot a full quality-vs-count curve to identify a saturation point?

4. Cross-identity reenactment with large appearance gaps:The paper shows convincing transfers across moderately different identities. I wonder if the authors tested situations where source and driver differ dramatically in gender, age, or ethnicity? If so, did expression fidelity or identity preservation degrade, and how might the method be extended to handle such large domain gaps more robustly?

---

### Official Review · Reviewer_WqQX · 2025-11-02

**Soundness:** 3
**Presentation:** 2
**Contribution:** 2
**Rating:** 2
**Confidence:** 4

**Summary:**

This paper present HIGH-Avatar, a one-shot 3D Gaussian head reconstruction framework using hierarchical representations. This method combines transformer-based global features and projection-based local features, fusing them with depth-guided occlusion awareness. It supports multi-level-of-detail modeling and separately reconstructs head and shoulders.

**Strengths:**

* This method balances detailed representation with computational efficiency through hierarchical subdivision and feature fusion.
* The approach of combining global features and local features, and utilizing depth buffers for occlusion-aware fusion, is reasonable and effective.

**Weaknesses:**

* The visualized results presented in this paper are difficult to capture the advantages of multi-level subdivisions.
* Modeling the head and shoulders separately did not demonstrate advantages. (refer to questions.)
* The paper lacks discussion and demonstration of limitations, extreme cases, and failure cases.
* As mentioned in the paper, the 3D Gaussian head model relies on FLAME priors and accurate 3D deformable model (3DMM) tracking, and it is difficult to capture subtle facial expressions.

**Questions:**

* In the visual results shown in Figure 3, the results of LAM in the 2nd, 3rd, 4th, and 9th rows, etc., show significant misalignment with the driven expressions and are worse than other baseline methods. However, in the quantitative results, the AED of LAM is still very high. What causes this problem?
* The detail advantages brought by multi-level subdivision are not obvious. For example, the white hair region in the third row of Figure 3, the beard region in the seventh row of Figure 3, and the scar on the head in the seventh row of Figure 12 indicate that the method may have difficulty capturing fine details.
* From Figure 2, the separate modeling of the shoulder region does not seem very reasonable. It has already been modeled by sub0/1/2. Considering that it is essentially modeled as a rigid body without additional controllability, is the modeling of sub0/1/2 already sufficient? At the same time, from the visualization results in Figure 3, the visual effect of the shoulder after separate processing is actually worse than some baselines (especially Portrait4Dv2).
* The results of the LAM method are missing in the speed comparison (Table 3).
* In the ablation experiment, using only the global feature leads to a severe loss of details. What would happen if only the local feature is used?
* From sub1 to sub2, although the number of Gaussian points increases significantly, the improvement in performance is very limited. What causes this?
* The results in Figure 3 show that the hair region in the reconstruction/driving results is worse than the baseline methods Portrait4Dv2/GAGAvatar (with smoother textures and loss of details). Why is there no improvement in the hair region after multi-level sampling?
* As can be seen from Figure 2, the predicted offset of the Gaussians in the hair region is clearly incorrect (even for the visible part of the frontal hair). What causes this? Will the same problem occur for frontal accessories such as glasses?

---

### Note · Authors · 2025-11-12

I have read and agree with the venue's withdrawal policy on behalf of myself and my co-authors.